# Support System for Etching Latte Art by Tracing Procedure Based on Projection Mapping

Momoka Kawai*
Tokyo Denki University

Shuhei Kodama†
Tokyo Denki University
ASTRODESIGN, Inc.

Tokiichiro Takahashi‡
Tokyo Denki University
ASTRODESIGN, Inc.

## ABSTRACT

It is difficult for beginners to create well-balanced etched latte art patterns using two fluids with different viscosities, such as foamed milk and syrup. However, it is not easy to create well-balanced etched latte art even while watching process videos that show procedures.

In this paper, we propose a system that supports beginners in creating well-balanced etched latte art by projecting the etching procedure directly onto a cappuccino.

In addition, we examine the similarity between etched latte art and design templates using background subtraction. The experimental results show the progress in creating well-balanced etched latte art using our system.

**Keywords:** Projection Mapping, Etching Latte Art, Learning Support System.

**Index Terms:** Human-centered computing—Human computer interaction (HCI)—User studies; Applied computing—Education—Computer-assisted instruction

## 1 INTRODUCTION

Etching latte art is the practice of drawing images on a coffee using a thin rod such as a toothpick [2]. There are several methods of etching latte art depending on tools and toppings. An easy method of etching latte art for beginners is pouring syrup directly onto milk foam and etching it to create patterns, as shown in Figure 1. The color combination obtained using syrup automatically makes a drink appear impressive. Hence, baristas are under less pressure to create a difficult design [8]. However, it is difficult for beginners to imagine how to pour syrup and etch it to create beautiful patterns because etching latte art involves two fluids with different viscosities. Furthermore, even though beginners can watch videos that show the procedure of etching latte art, latte art created through imitation does not appear well balanced. In this study, we define the etched latte art drawn in the middle of a coffee cup with unified lines as "well balanced." It is impossible to etch well-balanced latte art without repeated practice.

We develop a support system that helps even beginners etch well-balanced latte art by directly projecting the etching procedure using syrup onto a cappuccino. Moreover, the deformation of a fluid with viscosity, such as syrup, is projected through animations to support beginners in understanding the deformation of such fluids. We demonstrate the usefulness of this system through a questionnaire survey and examine the similarity between etched latte art and design templates using background subtraction. To the best of the authors' knowledge, there is no system that shows the procedure of etching latte art in the manner that our system does.

---

*e-mail: m-kawai@vcl.jp
†e-mail: s-kodama@vcl.jp
‡e-mail: toki@vcl.jp

## 2 RELATED WORK

### 2.1 Support for Creating Latte Art

There are two methods of creating latte art, etching and free pouring. In the former, syrup is poured on milk foam, and patterns are created via etching. In the latter, no tools are used and patterns are created using only the flow of milk, such as pouring.

Hu and Chi [4] proposed a simulation method that considered the viscosity of milk to express the flow of milk in latte art. The flow of milk obtained in their research was quite similar to that in actual latte art.

However, from the viewpoint of practicing latte art, users must estimate the paths of pouring milk and acquire the manipulation of a milk jug from simulation results. Moreover, it is difficult for users to understand the flow of milk unless they have advanced skills.

Pikalo [7] developed an automated latte art machine that used a modified inkjet cartridge to infuse tiny droplets of colorant into the upper layer of a beverage. Latte art with any design can be easily created using this machine, which is similar to a printer.

This machine enables everyone to create original latte art without any barista skills. However, the machine cannot create latte art using milk foam, such as free pour latte art. Therefore, baristas still require practice to create other kinds of latte art.

Kawai et al. [5] developed a free pour latte art support system that showed the procedure of pouring milk in the form of animated lines. This system targets baristas who have experience in creating basic free pour latte art and know the amount of milk to be poured. People without any experience in creating free pour latte art must practice several times.

### 2.2 Support for Creating Latte Art Using a Projector

Flagg et al. [3] developed a painting support system by projecting a painting procedure on a canvas. They placed two projectors behind users to prevent the projected painting procedure from being hidden by users' shadows. This system is quite large scale and expensive.

Morioka et al. [6] visually supported cooking by projecting how to chop ingredients at appropriate locations; this system can indicate how to chop ingredients and provide detailed notes and cooking procedures, which are difficult to understand during cooking while reading a recipe. However, users require a kitchen with a hole on the ceiling to use this system; this system is quite large scale because it projects instructions from a hole on the ceiling.

Xie et al. [9] proposed an interactive system that enables even inexperienced people to build large-scale balloon art in an easy and enjoyable manner using spatially augmented reality. This system provides fabrication guidance to illustrate the differences between the depth maps of a target three-dimensional shape and a work in progress. In addition, they designed a shaking animation for each number to increase user immersion.

Yoshida et al. [10] proposed an architecture-scale computer-assisted digital fabrication method that used a depth camera and projection mapping. This system captured a current work using a depth camera, compared a scanned geometry with a target shape, and then projected guiding information based on an evaluation. They mentioned that augmented reality devices, such as head-mounted

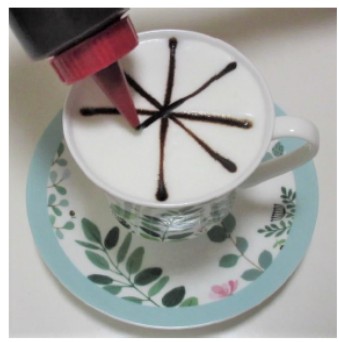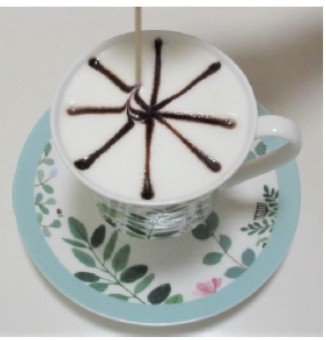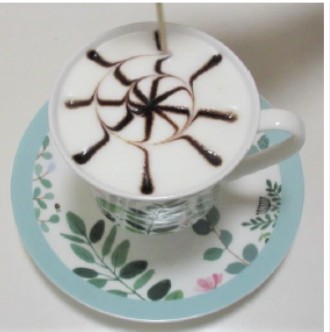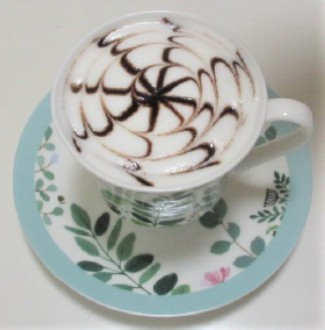

Figure 1: Procedure of etching latte art (left to right).

displays, could be used. However, the calibration of places displaying instructions for each device would be required. The proposed projector-camera guidance system can be prepared using a simple calibration process, and it allows for intuitive information sharing among workers.

As indicated in this paper, instructions can be projected on hands. Therefore, the difference between instructions and users' pick manipulation can be reduced. For this reason, we utilized a small projector to provide support for creating etched latte art.

## 3 PROPOSED METHOD

### 3.1 System Overview

The system overview is shown in Figure 2. The system configuration is provided below (Figure 3).

- A laptop computer is connected to a small projector.

- The projector shows a procedure on a cappuccino.

- The projector is placed on a tripod.

Firstly, users select a pattern among the Heart Ring, Leaf, or Spider Web patterns, as shown in Figure 2(a). Next, the users manually adjust the projector such that it points at a cappuccino. The procedure of creating the selected etched latte art is repeatedly projected on the cappuccino (Figure 2(b)). Then, the animation of syrup deformation is repeatedly projected on the cappuccino (Figure 2(c)).

Finally, the procedure is projected on the cappuccino once again. Users pour syrup to trace the projected image. Then, the latte art is completed via etching to trace the projected lines (Figure 2(d)).

The animations at each step can be played at any time.

### 3.2 Display Function of Procedure

In our system, users select a pattern from three kinds of etched latte art (the first column in Table 1). There are several other patterns of etched latte art as well. However, we adopted these patterns because they include all types of syrup placements (dots, curves, and straight lines) and pick manipulations (curves, straight lines, and spirals) used for creating etched latte art. A procedure is projected on the cappuccino after a pattern is selected.

First, the appropriate locations for placing syrup corresponding to the selected etched latte art are projected (the second column in Table 1). Next, the manipulation of the pick is projected (the third column in Table 1).

It is difficult to understand the procedure of etching latte art through books because they only show the sequence of several frames of a process video, as shown in Figure 1. Our system separately displays how to place syrup and manipulate the pick. As the

depth, angle, or speed of a tool does not significantly affect the final outcome, our system does not show any other information.

Table 1: Procedure of etching latte art using syrup.

| Etched Latte Art | Design Template | Syrup Place | Pick Manipulation |
|---|---|---|---|
| Heart Ring | | | |
| Leaf | | | |
| Spider Web | | | |

### 3.3 Animations of Syrup Deformation

It is difficult for beginners to imagine the procedure of syrup deformation through pick manipulation. Our system helps users understand this procedure by directly projecting prepared animations (Table 2). In our system, syrup is indicated in brown and the manipulation of the pick is indicated in blue. We used actual syrup and manipulated it using the pick to observe syrup deformation. Then, we created the animation to be the same as the actual syrup deformation by employing Adobe After Effects and considering two fluids with different viscosities. It required approximately 30 min to create each design template and 2 h to create each animation of syrup deformation.

### 3.4 System Configuration

As shown in Figure 3, a cappuccino is placed in front of a user and the projector mounted on the tripod is placed on the left side of the cappuccino (it is placed on the right side if the user is left handed). The procedure of etching latte art and the animations of syrup deformation are projected from the top.

## 4 EVALUATION

### 4.1 Evaluation Method

We conducted an experiment to evaluate our system. Twelve etched latte art beginners participated in the experiment. The participants

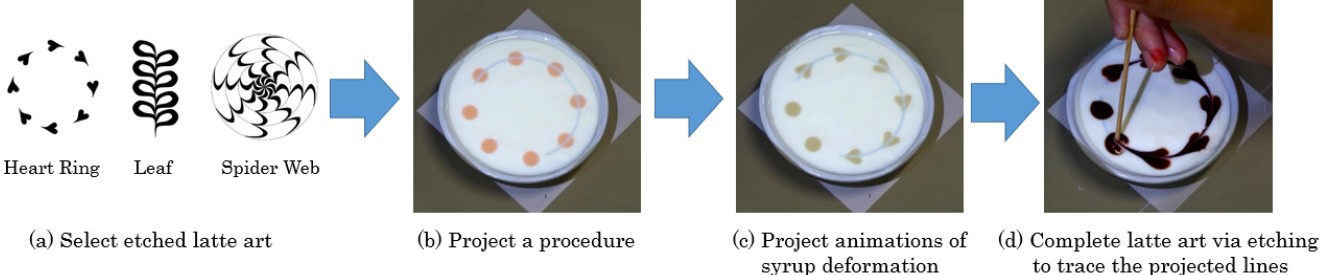

| Heart Ring | Leaf | Spider Web | | | |
|---|---|---|---|---|---|
| (a) Select etched latte art | | (b) Project a procedure | (c) Project animations of syrup deformation | (d) Complete latte art via etching to trace the projected lines | |

Figure 2: System overview.

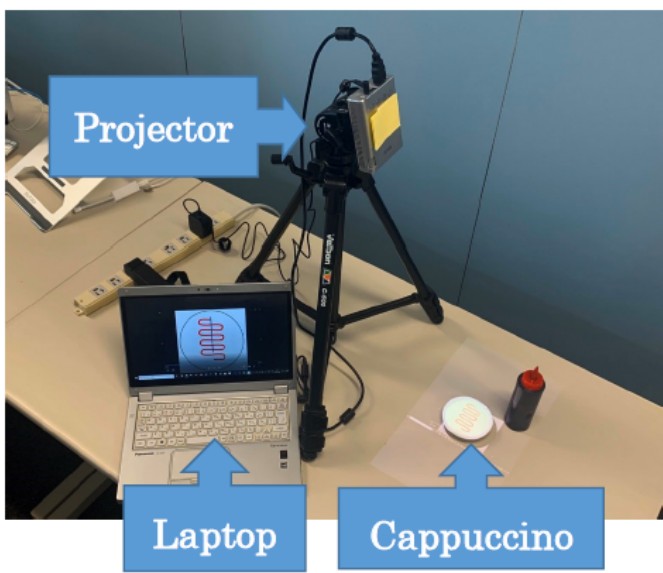

Figure 3: System configuration.

Table 2: Animations indicating the syrup deformation.

| Frame | 0% | 25% | 50% | 75% | 100% |
|---|---|---|---|---|---|
| Heart Ring | | | | | |
| Leaf | | | | | |
| Spider Web | | | | | |

Table 3: Process of etching latte art.

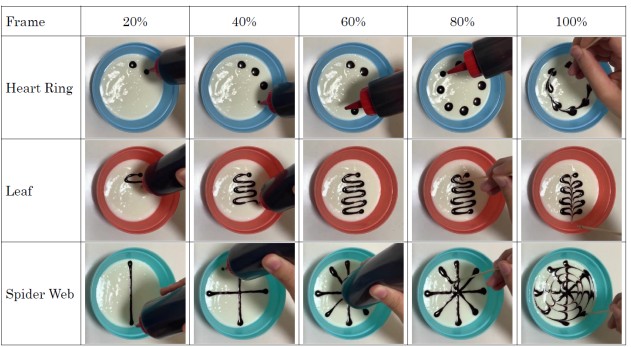

| Frame | 20% | 40% | 60% | 80% | 100% |
|---|---|---|---|---|---|
| Heart Ring | | | | | |
| Leaf | | | | | |
| Spider Web | | | | | |

were students in their 20s. They were not paid, and they did not aim to become baristas. Two of the participants had experience in drawing pictures (participants G and H ). We divided the participants into two groups (Group 1 and Group 2). To ensure homogeneity in the groups, the participants were randomly grouped and they never practiced creating etched latte art. The participants created etched latte art patterns using different methods (one without using the system and one with the system). Group 1 created etched latte art without using the system at first; thereafter, they created etched latte art using the system. In contrast, Group 2 created etched latte art using the system at first; thereafter, they created etched latte art without using the system. We selected patterns from the three previously mentioned etched latte art patterns so that all participants did not create the same pattern. The participants filled out a questionnaire after etching latte art.

Generally, in commercial use, baristas use well-steamed smooth foamed milk containing fair quality small-sized bubbles generated by a steamer attached to an espresso machine [1]. However, it is difficult to create such good quality foamed milk using a household milk frother. Milk foamed by a milk frother contains large bubbles that break easily; hence, syrup placed on this kind of milk foam spreads .

Therefore, in this experiment, participants created etched latte art using yoghurt. There is no difference between yoghurt and foamed milk in terms of the manipulation of the pick. Thus, we considered that using yoghurt in the experiment would not affect the evaluation of the system. We did not have to consider the difference resulting from using our system with a cappuccino .

(1) Etching Latte Art without System

Participants repeatedly watched a process video of etching latte art (Table 3). Then, they created etched latte art without using the system while watching the process video.

(2) Etching Latte Art with System

Participants created etched latte art using our system. First, they watched a procedure projected on the cappuccino. Second, the computer graphics animations of syrup deformation (animation speed

Table 4: Experimental result. Group 1 created etched latte art without using the system, whereas Group 2 created the art using our system. The red lines highlight the undesirable parts mentioned in Subsection 4.2.

was almost the same as the speed of actually etching latte art) were projected. Finally, the procedure of etching latte art was projected on the cappuccino once again, and the participants created etched latte art by tracing the projected procedure. The system could advance from the syrup placement to pick manipulation steps at any time. The experiment required approximately 60 to 100 s for each participant.

## 4.2 Results of Creating Etched Latte Art

The latte art etched by the participants is shown in Table 4. We compare and evaluate the etched latte art created without (row indicated by "Without system' in Table 4) and with our system (row indicated by "With system " in Table 4).

Participants A, B, G, and H created the "Heart Ring" pattern (Table 4 A, B, G, H). Participants B and H used an excessive amount of syrup when they did not use the system. Therefore, the hearts were extremely large and their shape was not as required. However, the participants were able to adjust the amount of syrup when they used the system. As a result, the shape of each heart was clearer and the quality of the etched latte art was better.

Participants C, D, I, and J created the "Leaf" pattern (Table 4 C, D, I, J). Participants C and J could not draw a line vertically. Participants J could not provide the same distance between each syrup. Their etched latte art appeared distorted because of these problems. They could create a well-balanced etched latte art with the same distance between each syrup using our system.

Participants E, F, K, and L created the "Spider Web" pattern

(Table 4 E, F, K, L). Participants E, F, and K could not draw a spiral within a certain space when they did not use the system. However, they could create the spiral using the system and produced better-balanced etched latte art.

Thus, the etched latte art created using our system was of good quality. It was therefore evidenced that even beginners could create well-balanced etched latte art using our system.

## 4.3 Participants' Questionnaire

Participants compared the etched latte art created with and without using the system. We conducted a questionnaire survey. The questionnaire consisted of the following five questions:

Question 1. Can you imagine how to create etched latte art before watching the process video?

Question 2. Is it easy to create etched latte art while watching the process video?

Question 3. Can you understand the syrup deformation from the animations projected by the system?

Question 4. Is the animation speed of syrup deformation appropriate?

Question 5. Is it easy to create etched latte art by tracing the procedure projected by the system?

The participants answered these questions on a 5-pt Likert scale (5: Yes, 4: Maybe, 3: Not confident, 2: Not too much, 1: Not at all).

In addition, the participants provided feedback on the system and the areas of improvement for the system.

Table 5: Results of questionnaire survey.

| | Questions | 1pt | 2pt | 3pt | 4pt | 5pt | Median |
|---|---|---|---|---|---|---|---|
| 1 | I can imagine how to create etched latte art before watching the process video. | 4 | 6 | 1 | 1 | 0 | 2 |
| 2 | It is easy to create etched latte art while watching the process video. | 1 | 5 | 1 | 3 | 2 | 2.5 |
| 3 | I can understand the syrup deformation from the animations projected by the system. | 0 | 0 | 0 | 2 | 10 | 5 |
| 4 | The animation speed of syrup deformation is appropriate. | 0 | 0 | 1 | 4 | 7 | 5 |
| 5 | It is easy to create etched latte art by tracing the procedure projected by the system. | 0 | 0 | 0 | 4 | 8 | 5 |

### 4.3.1 Questionnaire Survey Results

The results of the questionnaire survey are shown in Table 5.

Over 80 percent of the participants answered Question 1 as 1 pt or 2 pt, and the median was 2 pt, which is low. According to this result, etched latte art patterns are complex for people who see them for the first time. It is difficult for them to imagine how to create these patterns.

For Question 2, five participants answered 2 pt and only two participants answered 5 pt, and the median was 2.5 pt.

More than 90 percent of the participants answered questions 3 and 4 as 4 pt or 5 pt, and the median was 5 pt. We consider that the animations of syrup deformation provided by our system help users appropriately understand how syrup deforms when it is manipulated by the pick. In addition, the animation speed of syrup deformation is ideal.

All participants answered Question 5 as 4 pt or 5 pt. We consider that it is possible to create good quality etched latte art using our system. Our system is popular with participants because it projects the procedure directly on a cappuccino. Hence, the participants are not required to watch another screen that displays process videos while etching latte art.

### 4.3.2 Participants' Comments

The participants provided the following comments about creating etched latte art without using the system:

(1) I could not understand where to place the syrup because it was difficult to understand the location and the amount of syrup from the process video.

(2) It was difficult to manipulate the pick, and I could not create the desired pattern.

The participants provided the following comments about creating etched latte art using the system:

(3) It was easy to draw a line using the pick because I was only required to trace a line projected on the cappuccino. Thus, it was clear where and how much syrup I should place.

(4) I was delighted that I could create etched latte art even though I had never attempted it because the procedure was easy to understand.

(5) The animation of syrup deformation indicated how the syrup deforms. Therefore, I could imagine it.

The participants provided the following points of improvement for the system:

(6) I might have been able to place the suitable amount of syrup if the animations showed how to place syrup.

(7) At certain times, I found it difficult to trace lines from left to right because of my shadow.

(8) I was slightly confused because a large amount of syrup remained at the center and the line projected on the syrup was not visible.

### 4.3.3 Discussion

Based on the results in Table 4 and comments (1) to (5), we can conclude that even beginners are able to easily create etched latte art by utilizing our system as compared to watching process videos on another screen. This is because the system directly projects a procedure on a cappuccino, hence its popularity with users.

Users adjust the amount of the syrup by placing it on the pattern projected on the cappuccino. However, this pattern is a static image; hence, a few participants use an excessive amount of syrup, as stated in comment (6). We consider that preparing animations that show how to place syrup helps users clearly understand and imagine the speed and amount of syrup.

Additionally, as stated in comments (7) and (8), the projected procedure is difficult to view in a few cases owing to the position of users' hands or the color of the background. We will resolve this problem using multiple projectors.

## 4.4 Evaluation of Experimental Result

The purpose of latte art is to make customers enjoy not only the taste of coffee but also its appearance. Therefore, we consider that it is important for latte art to have a good appearance. Therefore, we performed a questionnaire survey among inexperienced people to determine which etched latte art (made without or with our system) appears more well-balanced, for the etched latte art created by each participant. Moreover, to obtain quantitative values of the etched latte art's appearance as aid of the inexperienced people's questionnaire result, we created foreground images through background subtraction for each design template and etched latte art to show which etched latte art is more similar to each design template.

### 4.4.1 Questionnaire Survey among Inexperienced People

Sixty inexperienced people who had not participated in the experiment were asked to determine the etched latte art (created without or with the system) that was more similar to the design template. The participants were the authors' acquaintances. Their ages ranged

from 19 to 58 years old. They were not paid for participation in the study, and two people had experience in creating free pour latte art.

The information about whether the etched latte art was created without or with the system was not provided to the participants.

The results of the questionnaire survey are shown in Figure 4.

The etched latte art created by ten participants out of twelve was more similar to the design template when the system was used compared to when the system was not used.

Participant G placed the appropriate amount of syrup at the suitable location even without the system. In this case, the etched latte art created without and with the system was well balanced.

Participant I could not follow the procedure appropriately because he/she placed syrup too rapidly. This participant stated that it might have been easier to create well-balanced etched latte art if the animations had shown how to place syrup. We will improve the system to resolve this issue by creating new animations that show the suitable speed of placing syrup.

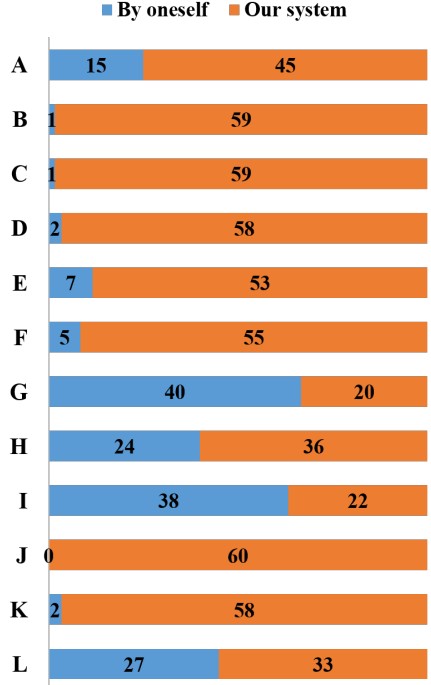

Figure 4: Result of questionnaire survey among inexperienced people. Sixty inexperienced people were asked which etched latte art (made without or with the system) looked more similar to the design template, for the latte art etched by each participant.

### 4.4.2 Background Subtraction

We created foreground images through background subtraction for each design template and etched latte art to quantitatively evaluate the etched latte art that was more similar to the design template. The white pixels in the foreground images indicated the difference between the design template and etched latte art, and black pixels indicated the parts that were the same.

We normalized the black pixels to quantify the similarity between each design template and etched latte art. A larger value indicated higher similarity.

The results of background subtraction are shown in Table 6.

The etched latte art created by ten participants out of twelve was more similar to the design template when the system was used compared to when the system was not used.

In the etched latte art created by Participant A, the location of each heart was adjusted by our system. However, the participant used an excessive amount of syrup. As a result, there was considerable difference between the etched latte art and design template.

In the etched latte art created by Participant D, the syrup was off to the right. As a result, there was significant difference between the etched latte art and design template. However, the difference between the etched latte art created without and with the system was only 0.1 percent, which was negligible.

We must indicate the appropriate amount of the syrup more clearly to obtain higher similarity between the design template and etched latte art.

### 4.4.3 Discussion

The results of the questionnaire survey of inexperienced people and background subtraction show that more than 80 percent of the participants created better-balanced etched latte art using our system. Two participants created better-balanced etched latte art without using the system. However, these participants were different in the cases of the questionnaire survey and background subtraction. Based on this result, we confirm there are instances where people consider that etched latte art is similar to the design template even though the result of the background subtraction shows that it is not, and vice versa. We must improve the system considering what kind of etched latte art people prefer.

In addition, we confirmed if the results for the participants who had experience in drawing pictures (participants G and H) were different from those of others.

As shown in Figure 4, the votes were divided among participants G and H. This may be because these participants created well-balanced etched latte art even without the system. However, the votes were also divided among participants I and L, who did not have experience in drawing pictures.

As shown in Table 6, the similarities between the design template and the etched latte art made without the system were 82.8 and 78.7 percent for participants G and H, respectively, whereas the similarity was 80.3 and 74.1 percent for participants A and B, respectively. No large difference was be confirmed between them.

Based on these results, we confirmed that the art backgrounds of the participants did not significantly affect the results.

In the future, we will confirm if the users make progress in creating better-balanced etched latte art by repeatedly using our system and if they can create well-balanced latte art even without the system.

## 5 CONCLUSION

We have developed a system that supports beginners in practicing and creating etched latte art and helps them understand syrup deformation by directly projecting the procedure and animations of syrup deformation onto a cappuccino. The participants' evaluations have verified the usefulness of our system.

As mentioned in Subsection 3.2, the procedure of our system is considerably simple. To the best of our knowledge, there is no system that shows the procedure of etching latte art in the manner that our system does. We will confirm the effect of simplifying the process video by simply demonstrating the procedure on a laptop.

The system has certain limitations, e.g., participants complained about the lack of information while using the system. Therefore, we will redesign the system by considering participants' views. Currently, our system is particularly effective for one-time use; however, the system might be used repeatedly by adding advice or correction functions and automatically creating animations of syrup deformation.

Moreover, we will examine the generalizability and scalability of our system. At present, our system supports only three patterns; however, other patterns can be supported by preparing design tem-

Table 6: Results of background subtraction. Similarities are represented by a number in the range of 0.000 to 1.000 (1.000 indicates that latte art is the same as the design template).

| | Participants | A | B | C | D | E | F |
|---|---|---|---|---|---|---|---|
| **Group 1** | Without system |  |  |  |  |  |  |
| | Similarity | 0.803 | 0.741 | 0.806 | 0.761 | 0.524 | 0.582 |
| | With system |  |  |  |  |  |  |
| | Similarity | 0.787 | 0.880 | 0.853 | 0.760 | 0.619 | 0.603 |
| | Participants | G | H | I | J | K | L |
| **Group 2** | Without system |  |  |  |  |  |  |
| | Similarity | 0.828 | 0.787 | 0.799 | 0.797 | 0.556 | 0.551 |
| | With system |  |  |  |  |  |  |
| | Similarity | 0.846 | 0.850 | 0.824 | 0.830 | 0.652 | 0.598 |

plates and animations. In addition, we will consider the points of improvement obtained in the survey.

Our system only uses a small projector. Thus, it is easy to adjust the projector to point at a canvas. Our system can be applied for decorating cookies or cakes with icing, which use materials with different viscosities, by appropriately changing the viscosity in the animation of syrup deformation.

## ACKNOWLEDGMENTS

The authors would like to thank Mr. Shigeaki Suzuki and AS-TRODESIGN,Inc. for their kind support, the reviewers for their helpful comments, and Editage (www.editage.com) for English language editing.

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
