# OpenReview forum: "Support System for Etching Latte Art by Tracing Procedure Based on Projection Mapping"
_graphicsinterface.org/Graphics_Interface/2020/Conference — GI 2020_

### Official Review · AnonReviewer3 · 2020-01-02
**Small but interesting system, needs some framing and info rework but is pretty solid**

**Confidence:** 4
**Rating:** 6

**Review:**

This paper presents a system designed to help latte art novices create etched patterns and images in the surface of coffee foam. The authors implemented the system and performed a user study (n=12). Using the system seems to help participants create art that is more similar to the presented templates.

Overall, this paper seems to be reasonably well-executed, with good figure support throughout the text (very important for a visual task like this!) and a nice implementation of a creation support tool. I find myself with some questions in the end, detailed below, but I think the paper is just about at the bar.

Goal of system: I am not clear if this system is intended to teach people how to create etching latte art unsupported, or if it is for constant use by a barista. The care that went into simulation of the two different liquid viscosities suggests that part of the goal may be to give intuition to the baristas about how the syrup and milk foam might interact, and the authors discuss in the intro that an earlier system which printed onto lattes “cannot make latte art with milk foam like one in free pour latte art. Therefore, baristas still have to practice latte art to make other kinds of latte art”, suggesting that their system could be used for bootstrapping knowledge into other kinds of latte art. However, the study only had users create two pieces (one supported and one unsupported, counterbalanced), and did not examine learning effects between the two conditions. On the same note, I find the simulation aspect of this work very interesting: I wonder how users would have done if they had been presented with a third condition that was a non-projection video of the etching creation (i.e., I wonder what effect the co-location of instruction and execution has vs. the effect of just simplifying the instructional video).

Mechanics of etching latte art creation: As a non-expert in latte art, I am curious about the other variables that go into creating successful etchings. Does the depth, angle, or speed of the tool affect the final outcome? Were other variables like these controlled or measured in the user study? I am also curious about the definition of “well-balanced” latte art. The authors use this term frequently throughout the text, but I don’t know what exactly it means.

System implementation: The authors describe the use of a small projector for displaying (and include a figure of it), but I am not clear how its video is projected onto the latte surface. Is there a depth camera component? Or is the projector simply adjusted to point at the latte each time? Does the animation loop over and over until the user completes the etching task, or what happens if a user misses the animation component of the training? I also do not completely understand how the fluid viscosities are simulated. Are there values that are plugged into After Effects to allow it to create this simulation? If so, what were the values, and how were they chosen? If it isn’t as simple as putting a viscosity value in for each fluid, how was this simulation done? The authors describe that making a template for this system takes about 30 minutes. What does a template author need to do or know in order to create templates for this system?

Study: I wish I knew a bit more about the study participants. How old were they? Were any of them aspiring baristas? What did they get for participating in the study (free coffee? money? extra credit?)? Did any of them have art backgrounds or other experience that might make their results different from others? I also think it would be beneficial to reformat the questionnaire results into a table or figure (for example, a box and whisker plot showing the distribution of answers on each question) to aid in exposition of that section. I like the two forms of post-hoc evaluation used for this study (both having humans look at the designs and doing background subtraction for creating a “more numerical” baseline of similarity. The fact that the background subtraction is so sensitive to shifting the design slightly to the side is also interesting.

Overall: As I said, I like this paper in general and find the topic pretty fun. While there are some lingering questions, they could be cleaned up by another revision pass (I would also suggest to the authors that they get a native English speaker to thoroughly proofread the paper). As I said, I think this paper is just about at the bar.

Nitpicking: I think in Table 1 the spiderweb design template is actually created using concentric circles, rather than a spiral (as the table seems to suggest). The spiderweb design elsewhere is correct. Structurally, it doesn’t make a lot of sense to have “making by oneself”, “making with our system”, and “notes as making etching latte art” as parallel pieces; I would bump “notes on making etching latte art” to a different level, since it is somewhat unrelated to the other two.

---

### Official Review · AnonReviewer1 · 2020-01-06
**Interesting system and study missing some details but good overall**

**Confidence:** 4
**Rating:** 6

**Review:**

This submission describes a projector-based system to help non-barista’s create latte art. The design of the system is presented, along with the results of a user study. The study found that most creations made with the system were more visually similar to templates than those created without the use of the system (as determined using image processing and questionnaire data from the public). Participants also found it easier to create latte art when using the projected system animations compared to watching a video of the process. The contribution of this submission comes from the system itself and from the results of the user study. It fits within the scope of the call for papers for GI 2020.

Overall, I enjoyed reading this submission and found that the domain of latte art presented some new opportunities for exploration that are not typically found within the skill acquisition literature. I also appreciated that the study included data not only from the participants but also from (what I assume to be) the general public’s opinions of the resulting latte art creations and a computer vision-based comparison. While the submission is hard to read in some places and some details about the system and study are missing, I think it is above the bar and should be accepted.


Readability: If possible, it would be for the submission to be copy-edited. There are quite a few places where it was difficult to understand what the intention of a paragraph or sentence was so the messaging was lost (e.g., “Even though making etching latte art while watching making videos which show the procedure, it is difficult to keep balance.” -> “Although one can watch a video that shows how to make etching latte art, videos make it difficult to maintain a learn how to balance different two fluids.”).

I also found the organization and presentation of the results to be confusing. In Section 4.2, to understand the text, one needs to reference Table 4, but Table 4 is found two pages later and it is unclear what the difference between the four sub-figures are. Moving the table closer to the text would provide clarity and adding annotations to the figures in the table would help call out the specific features of the figures that the text is referring to (i.e., I was not able to understand this description by looking at the figures in the table because they all appear to be very similar: “Participants E, F, and K were not able to draw a spiral with the certain space.”). I also found Section 4.3 difficult to understand because participant quotations, Likert ratings, and the result synthesis use some odd paragraphing and don’t provide clear takeaway messages. Perhaps organizing the results thematically, with one theme or main result per paragraph would improve clarity. Lastly when looking at Figure 4 without the text, it is unclear what questionnaire question was asked. Changing the caption to include the question would improve readability – it might also be a good idea to shrink this Figure because it’s a bit too large and takes up space that could be allotted towards describing the results in more detail.


Missing Details: The submission would be strengthened if there were details about the inexperienced people’s questionnaire process (e.g., where were they recruited from, how old were they, did they have experience with latte art, were they paid, etc.) and the participants in the study (same questions were lingering about experience, recruitment, age, payment, etc.). I was also unclear how long the study took to complete (i.e., length of time making each design), if participants were able to replay animations / the video while they were creating their designs, and how the system advanced from the syrup placement to pick manipulation steps.

More generally, after reading the submission I was left wondering about the generalizability of the system and what the unique challenges are to learners when they are learning to control and use fluids or gels. The Introduction touches a bit on this, but I was left wanting more of a discussion about the unique aspects of this domain in the Introduction, System Design, and possibly a new Discussion section about this.  One unique challenge seems to be that there is no room for error or mechanism to correct mistakes – if a design is messed up, the entire drink needs to be created again. This differs from other skill acquisition tasks where one could restart a task or remove what they have added. Because latte art is a form of food decoration, I was also wondering how the system could be appropriated or applied to decorating cookies or cakes with icing, which also use materials of different viscosity for decoration but don’t have the added element of heat / cooling. I encourage the author(s) to consider adding a Discussion section that highlights the findings of their study and discusses the generalizability of their work.


Lastly, for the Likert scale data, because Likert scales are ordinal rather than interval, the median instead of the average should be provided.

---

### Official Review · AnonReviewer2 · 2020-01-08
**Interesting**

**Confidence:** 3
**Rating:** 5

**Review:**

This paper presents a projection system to help unexperienced people to draw latte art on a cappuccino. There is a user study comparing participants performance with the system, and with watching explanatory videos only. The results suggest that participants perform better with the system.

This is overall an interesting idea of interactive system supporting skill acquisition. The system remains simple. This will not be a revolution, but it might be of interest.

To begin with, there is little details about the design rationale. What are the design choices? The system does not seem to follow a particular rationale. The fact that participants complained about the lack of information about syrup pouring reveals that this is more a trial and error approach than an informed design procedure.

There is no clue about scalability neither. To which extent the system supports other patterns? For example between the hears and the leaf the syrup is either a series of dots or a continuous line. This inevitably has an effect on syrup pouring. Are there other patters with features not presented in these three?

Looking at table 1 makes me think these instructions are quite clear on how to make these 3 patterns. I wish there was a condition with these schematics only. But it also makes me think about the actual difficulty of performing such art (I never tried myself). I expected more discussion on this point in the paper. It would have been a good start for a design rationale.

The experiment procedure give little details about participants background. How did authors ensure homogeneity of the groups?

Last, I would like to talk about the results. First of all I am unsure a pixel comparison metric is fair. The projection method inevitably show the precise spot for pouring syrup. But in the other condition, participants could have perform just as well, with a slight rotation or translation. This might have affected the metric, with no real impact on the perceived result. The discussion mentions participants who felt the drawing were similar while the metric showed they were not. What is the objective: people's perception or a metric? Also, how many times could participants practice? The results presented in appendix do not seem so different, and I think the result will be even more similar with a little practice.

In summary, the idea is interesting, but the design rationale is unclear, and it is unclear the results justify using this system.

---

### Meta-Review · Area_Chair1 · 2020-01-09

**Recommendation:** Accept
**Confidence:** 4

**Metareview:**

All of the reviewers found that the paper explores an interesting topic (latte art) although there are details about the study participants, scaling of the system, and system design choices that should be added to improve clarity. It would be beneficial for the author(s) to include such details and make the explanation of their target audience much clearer. Overall, it seems to be just above the bar and good enough for acceptance at GI.

---

### Decision · Program_Chairs · 2020-01-11

Accept